

# Estimating relationships between snow water equivalent, snow covered area, and topography to extend the Airborne Snow Observatory dataset

Dominik Schneider[1,2], Noah P. Molotch[1,2,3,*], Jeffrey S. Deems[4]

Correspondence to: noah.molotch@colorado.edu

[1]Dept. of Geography, Guggenheim 110, 260 UCB, Boulder Colorado 80309, USA
[2]Institute of Arctic and Alpine Research, University of Colorado Campus Box 450, Boulder, CO 80309, USA
[3]Jet Propulsion Laboratory, California Institute of Technology, 4800 Oak Grove Drive, Pasadena, CA 91109, USA
[4]National Snow and Ice Data Center, CIRES, 449 UCB, University of Colorado, Boulder, CO, 80309, USA

**Abstract**

A new spatio-temporal dataset from the ongoing Airborne Snow Observatory (ASO) provides an unprecedented look at the spatial and temporal patterns of snow water equivalent (SWE) over multiple years in the Tuolumne Basin, California, USA. We found that fractional snow covered area (fSCA) significantly improves our ability to model the distribution of SWE based on relationships between SWE, fSCA, and topography. Further, the broad availability of satellite images of fSCA facilitates the transfer of these relationship to different years with minimal degradation in performance ($r^2$=0.85, %MAE=33%, %Bias=1%) compared with models fit on the same day, by considering variations in SWE depth as expressed by differences in fSCA between years. The crux of this proposition is in selecting the model to transfer. We offer a method with which to select a model from another year based on the similarity in SWE distribution at existing snow pillows in the area. Comparison of the best transferred predictions and the selected predictions results in a mild decrease in $r^2$ (0.02) and moderate increases in %MAE (15%) and %Bias (10%). The results motivate further refinement in the technique used to select the best model because if these dates can be identified then SWE can be modeled at accuracy levels equivalent to models generated from ASO data collected on the day of interest. Lastly, we found that models from ASO observations of anomalously low snowpacks in 2015 still transferred to other years, although the same cannot be said for the reverse. This research provides a first attempt at extending the value of ASO beyond the observations and we expect ASO will continue to provide insights for improving water resource management for years to come.



## 1    Introduction

The spatial distribution of snow water equivalent (SWE) largely controls the timing and magnitude of streamflow (Stewart et al., 2004) and is important for plant ecology (Litaor et al., 2008) in snow-dominated catchments around the world. In these catchments, accurate assessments of snowpack water storage are critical for ensuring robust estimates of seasonal water supply. Nevertheless, SWE is poorly measured operationally with only sparse point measurements on the order of one measurement location per thousand square kilometers of snow covered terrain. Additional manual measurements in the form of snow courses add little information about the spatial variability and typically occur on the order of only three times a season.

Satellite remote sensing provides spatially explicit information about the snowpack but current satellites are unable to measure SWE directly at the scales relevant for water resources management in mountainous terrain (e.g. the western United States) (Dozier, 2011). The need for improved information regarding the quantity and distribution of SWE has led to the development of new measuring techniques including the application of ground penetrating radar (GPR) (Marshall and Koh, 2008), Global Positioning Systems (GPS) (Gutmann et al., 2012; Koch et al., 2014), Light Detection and Ranging (lidar) (Deems et al., 2013; Schirmer et al., 2011), and photogrammetry (Bühler et al., 2015; Nolan et al., 2015). Although these techniques are typically limited to snow depth, they can still capture the majority of the variability in SWE because snow depth varies an order of magnitude greater than density (Mizukami and Perica, 2008). Snow depth distributions can be converted to SWE using modeled snow density or in situ snow pit observations (Elder et al., 1991; Painter et al., 2016; Sturm et al., 2010).

Airborne systems have significantly improved the ability to measure SWE distribution at high spatial resolution and at extents relevant to water resource management (i.e. $> 100$ km$^2$). However, most previous studies of snow distribution using airborne data have been limited to snap-shots in time, limiting the ability to empirically transfer observations to time periods outside of those directly sampled. Since 2013, the National Aeronautics and Space Administration (NASA), Jet Propulsion Laboratory, Airborne Snow Observatory (ASO) has acquired weekly observations of snow properties from approximately the time of annual peak SWE to the end of the snowmelt season in the Tuolumne Basin, California (Painter et al., 2016). ASO measures snow depth by differencing the lidar-derived surfaces from a snow-on and snow-off flight and infers albedo and snow extent based on spectroradiometric measurements. An energy-balance model is used to estimate snowpack density, and subsequently convert snow depth to SWE. These weekly observations of SWE distribution over multiple years represent a new opportunity for understanding the spatial and temporal dynamics of snow distribution. Moreover, the intensive repeat sampling of ASO may provide an opportunity to extend observed SWE patterns beyond the time periods directly observed by ASO – a goal of the work presented here. In this context, the ability to extend expensive ASO data in time could dramatically expand the applicability of ASO data to future time periods without incurring the costs associated with future airborne acquisitions.

A potential application of ASO measurements relates to the possibility of developing statistical relationships between ASO data and other snow and terrain data that are more routinely available. In this context, statistical models have been extensively used to estimate relationships between snow point measurements and physiography (Balk and Elder, 2000; Elder et al., 1998; Erickson et al., 2005; Fassnacht et al., 2003; López-Moreno and Nogués-Bravo, 2006;





Molotch et al., 2005; Schneider and Molotch, 2016). At small scales (i.e. < 10 km$^2$), dedicated sampling of headwater
catchments has led to models that explain between 20% and 65% of the variability in snow depth based on
physiographic variables at 30m resolution (Balk and Elder, 2000; Elder et al., 1998; Erxleben et al., 2002).
Importantly, Erickson et al. (2005) found persistence in the topographic controls on snow depth distribution and
successfully parameterized a multi-year model relating the physiographic variables of a small headwater catchment to
annual peak snow depth by scaling the mean based on in situ measurements. Later studies have since confirmed an
inter-annual consistency in snow depth distribution based on high resolution lidar measurements (Deems et al., 2008;
Schirmer et al., 2011; Trujillo et al., 2009) and the inter-annual persistence of topographic controls on snow depth
distribution (Grünewald et al., 2013). These previous works suggest that relationships between lidar snow depth
measurements and topography may be useful for extending lidar measurements of snow in time.

Lidar is only capable of measuring snow depth and coincident density measurements are scarce, despite SWE

being the more important hydrologic variable. However, snow depth varies an order of magnitude more than density
and therefore largely controls the variability of SWE (Mizukami and Perica, 2008). Hence, we expect the results from
the previous studies be relevant here, i.e. inter-annual consistency in SWE distribution will be similar to that of snow
depth. Given the difficulty of extensively measuring density and SWE, operational SWE observation networks that
use snow pillows to measure SWE have been used to relate physiography and SWE. In this regard, multiple studies
have explained up to 82% of the variability in SWE based on physiographic variables (Fassnacht et al., 2003;
Harshburger et al., 2010; Schneider and Molotch, 2016). These studies aimed to understand the processes controlling
snow distribution and to apply this knowledge to interpolate point observations of SWE to >1000 km$^2$ for a single
point in time. The spatio-temporal dataset of SWE from ASO provides an unprecedented opportunity to develop
relationships between SWE and topography and test their persistence across several years.

Given that topographic variables are largely static in time, additional time-variant variables should be useful

in the context of explaining the spatio-temporal distribution of SWE. In this context, remotely sensed snow covered
area data has long been recognized to provide information with regard to snowpack water storage and consequently
expected summer streamflow (Good and Martinec, 1987; Martinec and Rango, 1981; Potts, 1937; 1944). Currently,
SCA is commonly estimated from SWE in hydrologic models through a depletion curve parameterization in order to
constrain melt production to the areal extent of snow cover (Anderson, 1973; Clark et al., 2011; Lawrence et al., 2011;
Livneh et al., 2010; Luce and Tarboton, 2004; Niu et al., 2011). The utility of depletion curves to provide sub-model-
scale information in physically-based modeling suggests that fSCA should provide additional information in statistical
models of SWE distribution. The consistent relationship between fSCA and SWE is predicated on the fact that SWE
distribution is extremely heterogeneous over complex terrain. Upon melt out, terrain features are progressively
uncovered. This process varies only slightly each year because of similarities in the meteorology, e.g. wind direction,
that drive accumulation patterns and solar exposure that drives melt out (Luce and Tarboton, 2004). Snow covered
area is also relatively easy to measure due to the distinctive spectral signature of snow compared to soil, rock, and
vegetation. In fact, photographic estimates of fSCA have been utilized for seven decades within hydrologic
applications (Parsons and Castle, 1959; Potts, 1937; 1944) and today robust observations of fSCA can be obtained
from variety of ground-based, aerial and satellite optical imagers (Bloschl et al., 1991; Dozier et al., 1981; Hall et al.,





2001; Kirnbauer and Bloschl, 1994; König and Sturm, 1998; Painter et al., 2009; Rittger et al., 2013; Rosenthal and
Dozier, 1996).
The repeat observations from ASO provide a unique dataset of concurrent SWE and fSCA over multiple
years with which to develop statistical relationships between SWE, fSCA, and topography. Given that fSCA is widely
observable from a variety of satellites, these relationships could then be used to estimate SWE for any date on which
fSCA observations are available. Hence, the objective of this research is to use ASO-derived relationships between
SWE (dependent variable), and fSCA and physiography (independent variables) to estimate SWE distribution for time
periods when ASO data are not available. We aim to test how well statistical models of the relationship between SWE,
fSCA and physiography transfer in time. We ask (1) Does fSCA improve statistical models of SWE distribution? (2)
Can statistical models of SWE distribution be transferred directly from one year to another? (3) How can we determine
which SWE distribution from the ASO record best represents a date of interest?
We present our SWE distribution modelling framework and show the utility of including the time-variant
variable fSCA for improving the SWE distribution estimates. Further, we evaluate the impact of transferring models
from one year to another. Lastly, we present a methodology for identifying which models of SWE distribution, from
the ensemble of historical ASO acquisitions, best represents the SWE distribution for unsampled dates of interest. We
then discuss the results in the context of extending ASO to unsampled dates.
**2    Site Description**
We used a SWE dataset from the Tuolumne River basin in the Sierra Nevada mountains in California, USA (Fig. 1).
The basin is 1,175 km$^2$ in area, consisting of 48% vegetation, 50% rock, 2% water, and small isolated areas with
permanent snow/ice. The elevation range is 1127 m to 3965 m, encompassing 4 distinct ecological zones ranging from
lower montane forest to alpine (NPS, 2016). The lower montane forest ranges from 1127 m to 1800 m elevation and
consists of a diverse mix of coniferous and deciduous trees. The upper montane forest ranges from 1800 m to 2450 m
elevation and primarily consists of coniferous species such as red fir and lodgepole pine. Elevations from 2450 m to
2900 m are considered subalpine and consist of a mix of meadows and coniferous forest. The highest elevation band
above 2900 m is an alpine zone that is devoid of tree cover and contains limited herbaceous vegetation. This alpine
zone contains areas with large granitic features, talus slopes and boulder fields. Snowmelt from the basin runs off into
the Hetch Hetchy reservoir, which is the main water supply for the City of San Francisco.






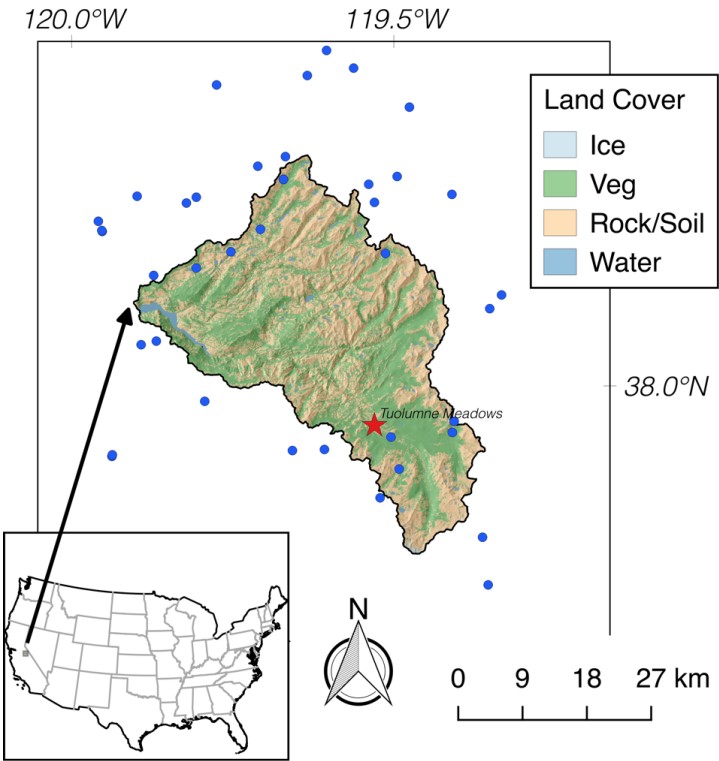


**Figure 1. Land cover map of the Tuolumne Basin with snow pillows shown as blue dots. The land cover data was provided**
**by ASO and was derived from spectral information from summer snow-off flight. The snow pillows locations were obtained**
**from the California Department of Water Resources.**

## 3    Data Sources

We utilize the ASO dataset in the Tuolumne River Basin in the Sierra Nevada mountains of California. The ASO
mission consists of airborne lidar in conjunction with a hyperspectral spectroradiometer (Painter et al., 2016). The
dataset consists of a 3 m resolution snow-free digital elevation model (DEM), 3 m snow depth maps for which snow-
free areas are masked using spectral information from the spectroradiometer, 3 m vegetation height map and 50 m
SWE maps. The dataset is distributed in the UTM zone 11 and WGS84 datum map projection system.

In order to conduct our analysis at a spatial scale that will ensure transferability to more widely available
data, we mean-aggregated the 50 m SWE maps to 500 m. Subsequently, we converted the 3 m DEM and snow depth
maps to 501 m using mean aggregation and then bilinearly resampled to the 500 m SWE grid. We computed the
physiographic variables used in the modelling framework [Table 1] from the 500 m DEM using open source GIS
software, including GDAL (GDAL Development Team, 2015), SAGA GIS (Conrad et al., 2015), and R (R Core
Team, 2015). We chose the 500 m resolution because it is relevant to water resource management and is the scale for
which daily satellite fSCA images are available from the Moderate Resolution Imaging Spectroradiometer, which is
a commonly used fSCA data product (Painter et al., 2009; Salomonson and Appel, 2004). We also used a binary



aggregation of the 3 m snow depth maps to obtain 501 m fSCA maps, which were then resampled to 500 m by nearest
neighbor to preserve snow free pixels. Lastly, we aggregated and bilinearly resampled the vegetation height map to
500 m.

**Table 1. List of physiographic variables considered in the multiple linear regression to model SWE distribution.  Source**
**includes studies in which these variables have been used and the source of the algorithm, if applicable. Citations in the**
**source column are by no means exclusive.**

| Variable | Units/Derivation Specifics | Source |
| --- | --- | --- |
| UTM Northing | meters | Fassnacht et al. (2003) |
| UTM Easting | meters | Fassnacht et al. (2003) |
| elevation | meters | Elder et al., (1991, 1998) |
| zness | sine(slope); ranges 0-1; dimensionless | Balk and Elder (2000); Erxleben et al. (2002); Fassnacht et al. (2003) |
| northness | cosine(aspect); ranges 0-1; dimensionless | Balk and Elder (2000); Erxleben et al. (2002); Fassnacht et al. (2003) |
| eastness | sine(aspect); ranges 0-1; dimensionless | Balk and Elder (2000); Erxleben et al. (2002); Fassnacht et al. (2003) |
| topographic position index (TPI) | elevation difference of a pixel from the mean of the surrounding pixels; meters | Revuelto et al. (2014); GDAL (2015) |
| vector ruggedness measure (VRM) | 3-dimensional measure of the variation of slope and aspect; not correlated with slope or aspect; ranges 0-1; dimensionless | Veitinger et al. (2015); Sappington et al. (2007); Conrad et al. (2015) |
| standard deviation of slope | standard deviation of slope in 3x3 window around each pixel; shown to detect changes in slope at multiple scales; radians | Marchand and Killingveit (2005); Lopez et al. (2014); Grohman et al. (2007) |
| vegetation height | measured by ASO; used in place of forest canopy density from previous studies; meters | Molotch and Bales (2005, 2006); Painter et al. (2016) |



The ASO flies approximately weekly from near peak SWE to the end of the melt season. As such, there were
six flights in 2013, ten in 2014, eight in 2015, and eight in 2016. Only the final four flights of the 2016 season were
available for this study resulting in a total of 28 SWE maps. The 2014 and 2015 seasons were characterized by a severe
dry snow drought (Harpold et al., 2017) and 2013 and 2016 also experienced below average snowpack conditions, but
less severely. Painter et al. (2016) report a mean absolute vertical snow depth error of 8 cm and a bias of 1 cm when
compared with manually measured snow depths at the 15 × 15 m scale. Further details about the mission and
processing can be found in Painter et al. (2016).
We also obtained daily SWE measurements from 54 snow pillows operated by the California Department of
Water Resources that are within 20 km of the Tuolumne watershed boundary. The stations range from 2000 m to 3250
m. We downloaded the adjusted SWE records, which have been manually quality controlled, for the 2013-2016 water
years. No further adjustments were performed. The data can be downloaded from http://cdec.water.ca.gov/.
**4    Methods**
We use linear regression to model the distribution of SWE for every ASO flight. The explanatory variables we consider
are ASO-observed fSCA and physiographic variables previously used in the literature (Table 1). We present results




from two models: 1. "PHV" is a multiple linear regression that consists only of physiographic variables as independent
variables. This represents the traditional approach for estimating SWE distribution based on relationships with
physiography; 2. "PHV-FSCA" is a multiple linear regression that includes both physiographic variables and fSCA
as independent variables. We make these distinctions to demonstrate the utility of fSCA as a time-variant variable. No
one has explicitly explored the utility of fSCA for directly estimating the distribution of SWE. fSCA values greater
than and equal to zero are used to fit the PHV-FSCA models. Regression estimates from both PHV and PHV-FSCA
are masked to snow covered areas as observed by ASO. Once we illustrate the utility of the statistical models for
characterizing ASO SWE distribution patterns at discrete points in time, we then show how these statistical models
can be transferred to time periods without ASO observations. With regard to all statistical models, we report the
squared Pearson correlation coefficient ($r^2$) as a measure of the relative spatial pattern between the modeled SWE
distribution and ASO observed SWE distribution. We also report the mean absolute error as a percent of mean
observed SWE (%MAE) as a measure of the accuracy of the modeled SWE distribution. Lastly, we report bias as a
percent of mean observed SWE (%Bias) as a measure of the systematic over or under-prediction by the model.

### 4.1    SWE Models: Discrete Time

To examine the utility of fSCA as a predictor, we compared the SWE distributions modeled with PHV and PHV-
FSCA using a split sampling strategy. We split each date into a training (80%) and test (20%) dataset to evaluate
overall model performance on this date. This insures that we are not evaluating the model with the same data used to
create the model. Furthermore, this procedure is replicated 20 times to provide 20 different subsets with which to
evaluate model performance; this is more robust than a single replication. More replications were computationally
prohibitive. This split sample strategy is an important initial step in transferring ASO data in time as it is necessary to
first show that fSCA and physiographic variables can be used to adequately model ASO-observed SWE on the date
of acquisition. Once this is established, the transferability of the models in time can then be explored – as described
in the next section.

### 4.2    SWE Models: Transferred in Time

We evaluated a second set of predictions whereby each date is modeled using all the data, i.e. not split, and then this
model is used to predict SWE on dates that ASO flew in different years. In this manner, we simulate SWE on the date
ASO flew using models from other years and then we use the ASO data on the date of interest strictly to evaluate the
model estimates of SWE. Hereafter, we refer to the date of the model (i.e. the date of the ASO observation for which
the model is developed) as the *model date* and the date being predicted as the *transfer date*. This results in 28 models
of SWE distribution for PHV and PHV-FSCA each because there are a total of 28 ASO flights. Given our primary
goal of estimating the SWE distribution for unsampled dates, we apply models developed for each ASO acquisition
to all other dates except for dates within the same year as that in which the model was developed. For example, in
2013 there were a total of 6 ASO flights out of the total of 28 flights during the four-year study period. This leaves 22
flights from other years that can be used to develop statistical models of SWE that can be transferred to the dates in
2013. By conducting our model tests in this manner we are more robustly testing the transferability of models from a



given *model date* with regard to simulating SWE distribution on a given *transfer date*. Due to a different number of
ASO observations in each year, the prediction ensemble size differs for each year. As noted above, for each date in
2013 there are 22 potential models. For each date in 2014 there are 18 potential models; for each date in 2015 there
are 20 potential models; and for each date in 2016 there are 24 potential models. These models are referred to as
*transferred models*, and for each transfer date we identify the best model from another year based on error statistics
generated from the ASO data acquired on the transfer date. We refer to this model as the *best model*.

The ability of a model to transfer from the one date to another will vary based on how well the relationships

between the dependent variable (SWE) and explanatory variables are captured and the similarity of the SWE
distributions. Here we quantify the SWE similarity between dates using the mean absolute error (MAE) of SWE
recorded at nearby snow pillows. For each transfer date, there is an ensemble of predictions from the model dates from
the other years. Each of the model dates exhibit a similarity with the SWE distribution of the transfer date. In order to
pick which model date exhibits the greatest similarity with the transfer date without having an ASO observation, we
calculate the MAE of SWE at the snow pillows between each pair of model-transfer dates and select the model date
with the lowest MAE. We compare the prediction performances from this model selection procedure (denoted *selected*
*model*) with those of the best models.
**4.3    Statistical Model**
The multiple linear regression models described above, i.e. PHV and PHV-FSCA, are based on a regularized
regression model applied in an elastic net framework as implemented in the glmnet package in R (Friedman et al.,
2010). The benefit of a regularized regression over standard regression is that it reduces overfitting while permitting
all conceptual variables to be included, rather than removing potentially useful variables due to multicollinearity.
Regularized regression increases the predictive ability of a model with multiple predictor variables by penalizing the
objective function used to estimate the parameter set. The elastic net is an extension of ordinary least squares, which
estimates parameter coefficients by minimizing the residual sum of squares (RSS) as the objective function (Eqn 1):

$$RSS = \sum_{i=1}^{n} \left( y_i - \sum_{j=1}^{p} \beta_j x_{ij} \right)^2 \qquad\qquad \text{Eqn (1)}$$


where $y_i$ is the response variable at the $i^{th}$ observation, $\beta_j$ is the coefficient for predictor variable j, and $x_{ij}$ is predictor
variable j at each observation i. The elastic net penalizes RSS by two different types of regularization techniques,
known as L1 and L2, that have opposing properties (more on this below). In this regard, the elastic net estimates the
regression parameters β by minimizing RSS in Eqn 2:

$$RSS = \sum_{i=1}^{n} \left( y_i - \beta_0 - \sum_{j=1}^{p} \beta_j x_{ij} \right)^2 + \lambda P_\alpha(\beta) \qquad\qquad \text{Eqn (2)}$$






where

$$P_\alpha(\beta) = \sum_{j=1}^{p} \left[ \frac{1}{2}(1-\alpha)\beta_j^2 + \alpha|\beta_j| \right] \qquad \text{Eqn (3)}$$


$\beta_0$ is the intercept, $\lambda$ controls the magnitude of the penalty, and $\alpha$ changes the relative influences of the L1 and L2
regularizations. In practice, this shrinks the coefficient values towards zero to account for multicollinearity and
predictors with low explanatory power. Penalized coefficients have less variance and can select variables without
resorting to a discrete selection procedure, e.g. a p-value threshold such as in step-wise regression. Consequently,
resulting parameter sets are more robust predictors for independent data (Zou and Hastie, 2005a).

The elastic net has two tuning parameters, $\lambda$ and $\alpha$, which are determined through cross validation. When

$\alpha$=1, the penalty is composed completely of the L1 penalty and commonly known as Lasso regression. When $\alpha$=0,
the penalty is composed completely of the L2 penalty and is commonly known as Ridge regression. The advantage of
the elastic net is that alpha can range between 0 and 1 and therefore inherits the properties of both L1 and L2
regularization. L1 regularization is commonly used for model selection because predictor coefficients can be shrunk
to zero and effectively removed from the model. However, in the presence of correlated predictor variables, one
predictor variable would be randomly selected while the others are removed. This can result in decreased predictive
performance since variables with some explanatory power are no longer in the model. With L2 regularization,
regression coefficients will shrink towards zero but with an asymptote at zero. This is the preferred type of
regularization in the presence of multicollinearity because all variables would remain in the model but with smaller
coefficients. The elastic net provides a framework to choose the best compromise between the L1 and L2 penalties.
For further details, we direct the reader to Zou and Hastie (2005b) and Hastie et al. (2009).

We do not directly treat spatial correlation in our models due to the large computational demands of fitting

the covariance function for ~4000 pixels for 28 dates. Neglecting spatial correlation is another potential source for
regression coefficients to be overfit and consequently we do not interpret them for physical meaning (Cressie, 1993;
Erickson et al., 2005). Nonetheless, we show utility with our methods without addressing spatial correlation and expect
the results presented herein would improve if spatial correlation were explicitly treated (Carroll and Cressie, 1997).
**5     Results**
**5.1     SWE Models: Discrete Time**
Table 2 shows that PHV-FSCA outperforms PHV in all metrics in all years except %Bias (where both models exhibit
close to 0 %Bias, as expected from a regression). The model PHV-FSCA explains on average between 78% and 86%
of the variance in SWE distribution in any given year whereas PHV only explains between 55% and 67%. We similarly
see improvement in %MAE where PHV-FSCA exhibits mean annual %MAE between 27% and 41% compared with



PHV which yields mean annual %MAE between 50% and 71%. In summary, we note a substantial improvement in $r^2$
and %MAE for distributing SWE when fSCA is included as an additional variable.

**Table 2. Mean prediction performance for PHV and PHV-FSCA from each observation date using a split-sampling**
**approach.**

| | PHV | | | PHV-FSCA | | |
|------|------|------|-------|------|------|-------|
| | $r^2$ | %MAE | %Bias | $r^2$ | %MAE | %Bias |
| 2013 | 0.55 | 71 | 0 | 0.86 | 33 | 1 |
| 2014 | 0.67 | 50 | 1 | 0.86 | 27 | 1 |
| 2015 | 0.57 | 64 | -3 | 0.83 | 33 | 1 |
| 2016 | 0.59 | 71 | 2 | 0.82 | 41 | 2 |
| **Mean** | **0.6** | **61** | **0** | **0.85** | **32** | **1** |



**5.2    SWE Models: Transferred in Time**
Figure 2 shows March 23, 2014 as an example transfer date in 2014 that was predicted using a PHV-FSCA model
from May 25, 2013. Overall, we see similar spatial trends between the observed and modeled SWE distributions, but
we see darker purples in the observed map indicating higher SWE. The mean observed SWE is 0.23 m compared to
0.21 m modeled. The range of observed SWE is 0-0.75 m while the modeled SWE ranges from 0-0.39 m. The standard
deviations of observed and modeled SWE are 0.13 m and 0.11 m, respectively. The difference map shows large areas
of agreement to within 0.05 m SWE and a qualitative comparison with Figure 1 suggests these are mainly forested
areas. We see areas of under prediction (red) mostly above tree line in the north and areas of over prediction (blue)
above tree line in the south. A comparison with Google Earth® aerial imagery confirms that the pixels that exhibit
very large negative differences (bright red pixels) are areas with persistent snow for much of the year. The snow extent
is very similar between the modeled SWE and observed SWE because only areas observed to have fSCA greater than
0 were predicted.





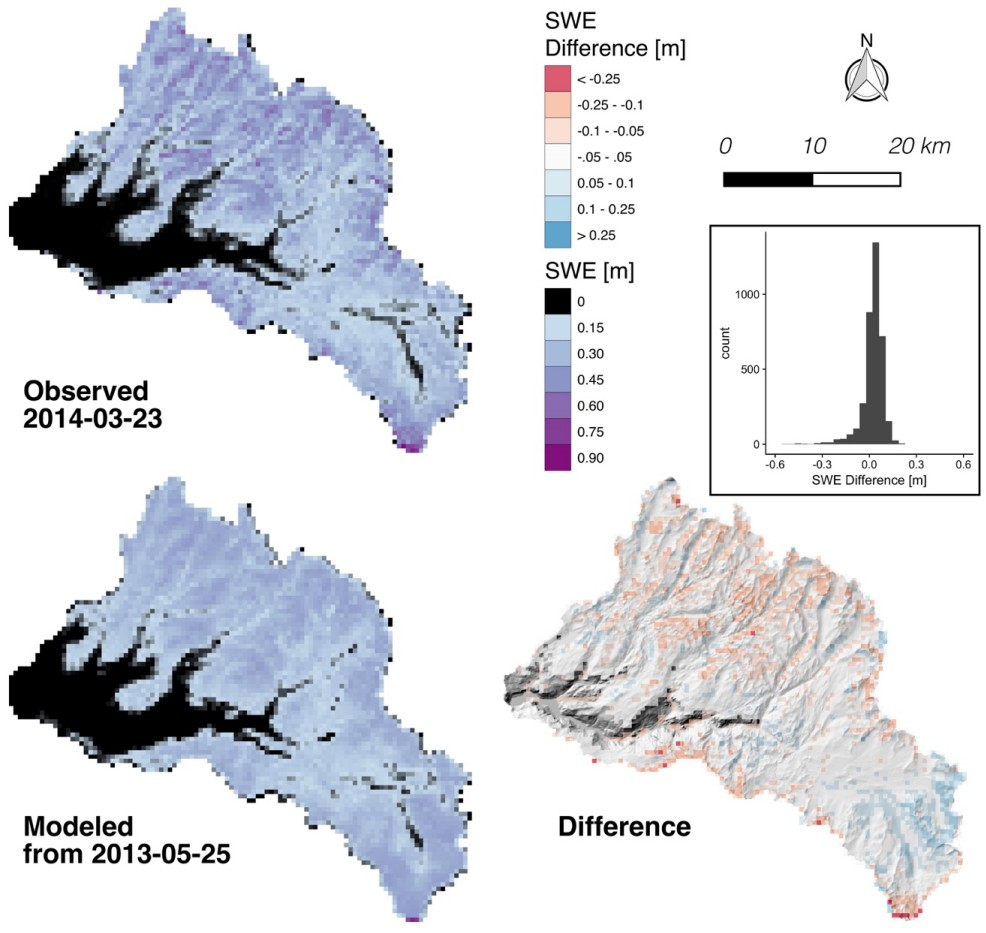


**Figure 2. An example of SWE distribution from 2014-03-23. Observed SWE (top left); modeled SWE using PHV-FSCA (bottom left); the difference, modeled SWE – observed SWE, draped over a shaded relief map (bottom right). A histogram of the differences (top right).**


Figure 3 shows the range of $r^2$ for transferred models for PHV and PHV-FSCA on each date based on models

created in other years. Additionally, we indicate the best model performance for each transfer date by a diamond. In
this regard, we observe unanimous improvement across all dates with PHV-FSCA compared to PHV. PHV-FSCA
yields the highest mean best $r^2$ of 0.84 (mean of diamonds) compared to PHV with a mean best $r^2$ of 0.6. We also
observe a notable decrease in $r^2$ for PHV towards the end of the season while PHV-FSCA exhibit a consistent $r^2$. The
standard deviation of $r^2$ for PHV is 0.14 and for PHV-FSCA it is 0.05.





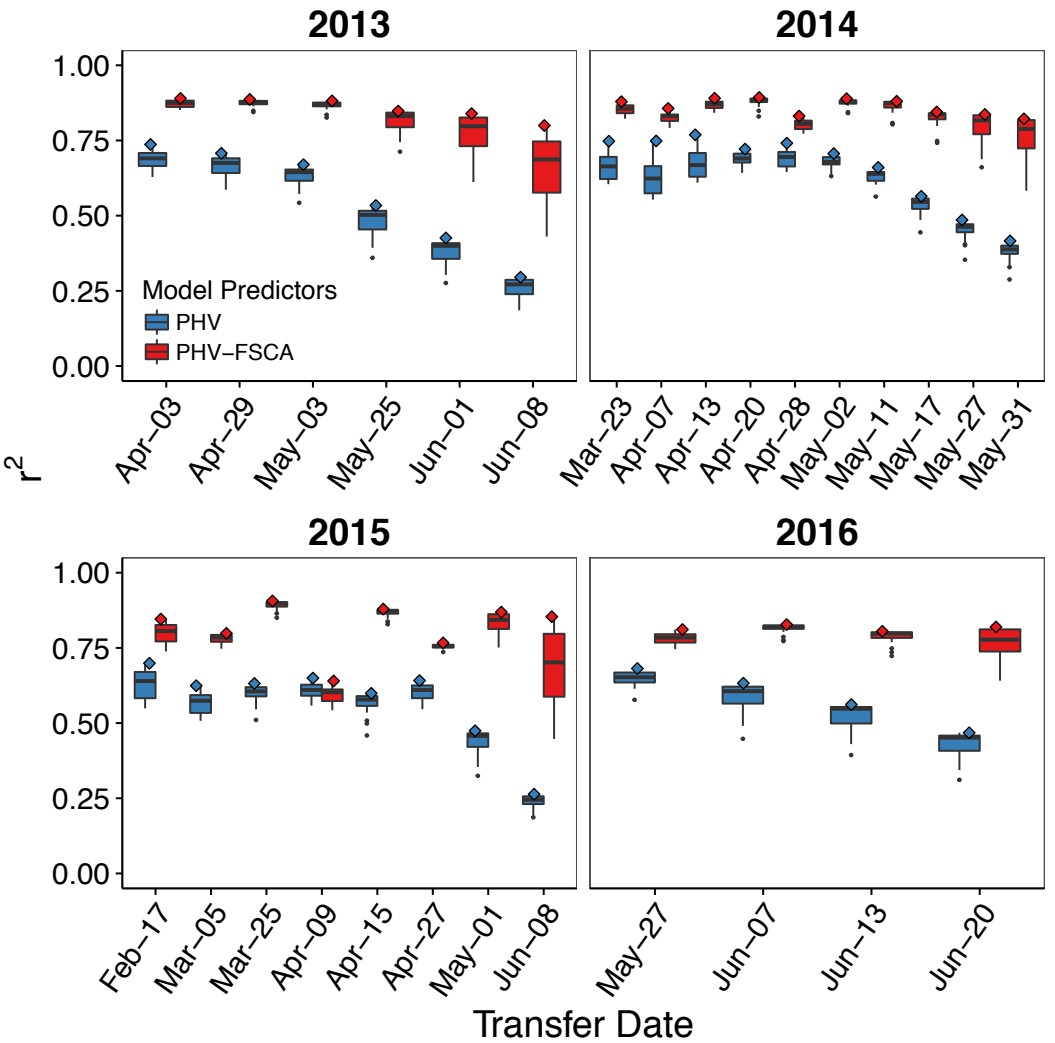

**Figure 3. The range of r$^2$ for each simulation date from the transferred models. The diamond represents the best model for each transfer date. The boxplots represent the interquartile range with vertical lines to denote the 5th and 95th percentiles. Black dots are outliers.**

Figure 4 also clearly shows PHV-FSCA to exhibit the best, i.e. lowest, %MAE with transferred models compared to PHV. The mean best %MAE (mean of diamonds) for PHV-FSCA is 33% while the mean best %MAE for PHV is 63%. Particularly obvious in these panels is the upward distribution shift and larger range for PHV later in the season compared to only a minimal increase in %MAE for PHV-FSCA. The standard deviations of the best %MAE are 10% for PHV-FSCA and 26% for PHV.




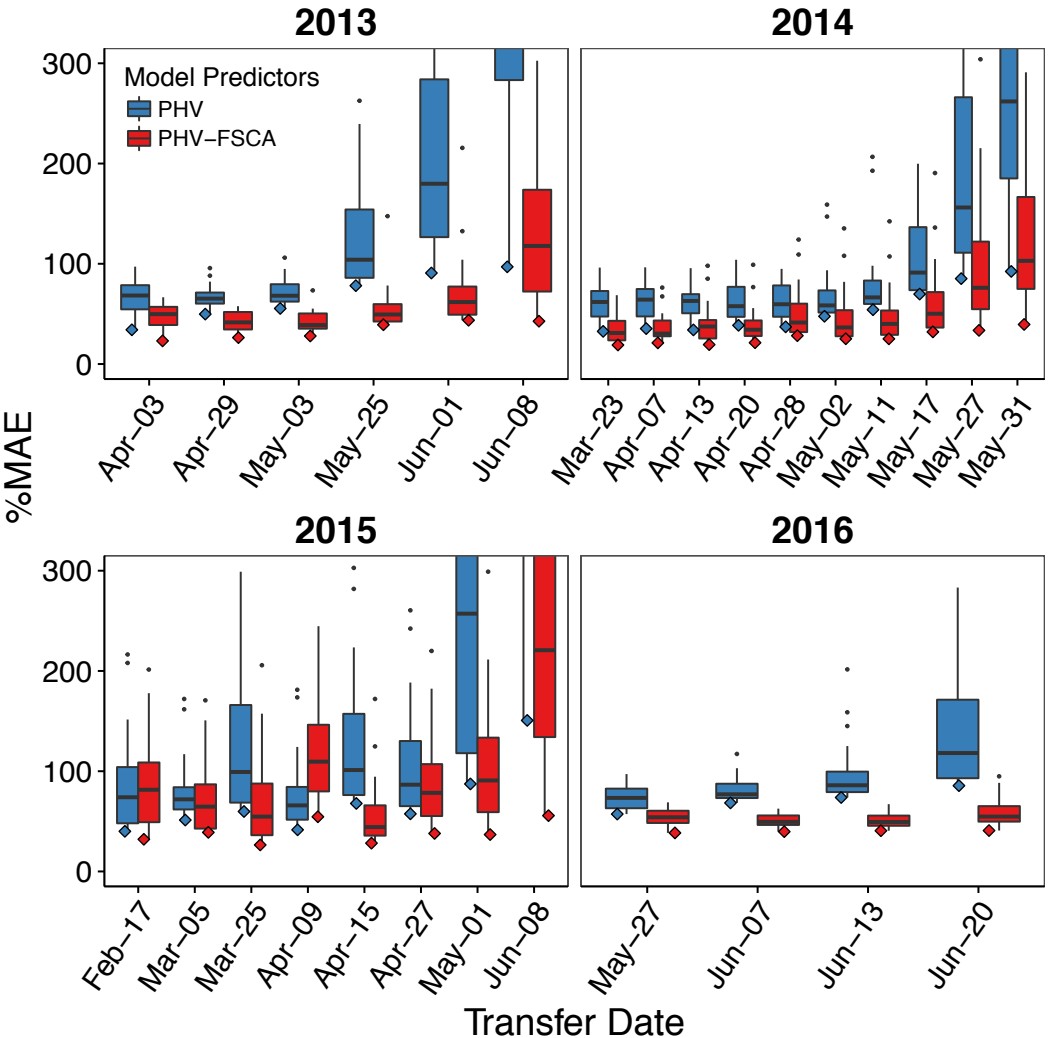

**Figure 4. The range of %MAE prediction errors for each transfer date from the transferred models. The diamond represents the best model for each transfer date. The y-axis is limited for clarity. The boxplots represent the interquartile range with vertical lines to denote the 5th and 95th percentiles. Black dots are outliers.**

Figure 5 shows that the best transferred models (diamonds) for both all models exhibit close to zero bias. The mean best %Bias for PHV is 2% and for PHV-FSCA it is 1%. However, we note that the variability in %Bias increases more dramatically at the end of the season, especially for PHV. The standard deviations of the best (i.e. lowest) %Bias for PHV and PHV-FSCA are 15% and 7%, respectively.



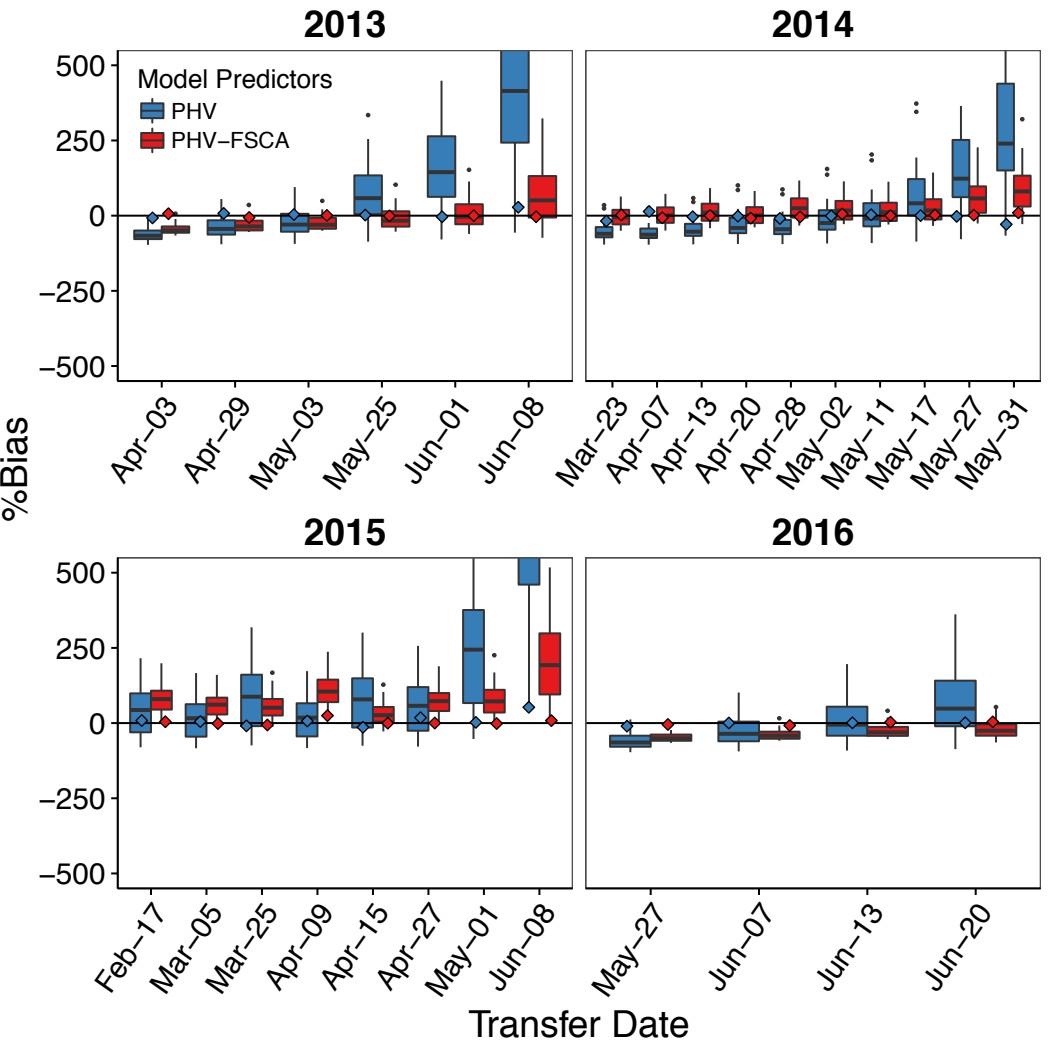

**Figure 5. The range of %Bias prediction errors for each transfer date from the transferred models. The diamond represents the best model for each transfer date. The y-axis is limited for clarity. The boxplots represent the interquartile range with vertical lines to denote the 5th and 95th percentiles. Black dots are outliers.**

Following the demonstration of unanimous improvement with fSCA as a predictor variable, we compare just the best transferred models of PHV-FSCA presented in Figs. 3, 4, 5 with the split-sample models of PHV-FSCA from the previous section, to assess the performance degradation one would expect due to transferring a model between years. We observe that PHV-FSCA models can be transferred to another year with little degradation in performance (Table 3). The yearly $r^2$ of the best transferred model are always within 1% of the split sample model and, on average, explains the same amount of variance in SWE distribution. The yearly mean %MAE of the transferred model is always within 6% of the split sample model with the mean 1% higher. The yearly mean magnitude of %Bias is actually lower,



i.e. better, in two years with the best transferred model compared to the split sample model, and the overall mean is
the same.

**Table 3. Mean model performance comparison with PHV-FSCA for the split sample model and the best transferred model**
**for each simulation date.**

|  | Split Sample Model | | | Best Transferred Model | | |
|---|---|---|---|---|---|---|
|  | $r^2$ | %MAE | %Bias | $r^2$ | %MAE | %Bias |
| 2013 | 0.86 | 33 | 1 | 0.86 | 34 | -1 |
| 2014 | 0.86 | 27 | 1 | 0.86 | 26 | 0 |
| 2015 | 0.83 | 33 | 1 | 0.82 | 39 | 3 |
| 2016 | 0.82 | 41 | 2 | 0.82 | 40 | -1 |
| **Mean** | **0.85** | **32** | **1** | **0.85** | **33** | **1** |


**5.3        Evaluating the Selected Model Performance**
Table 4 summarizes and compares the yearly mean statistics from PHV-FSCA for the best models and the selected
models transferred from another year. The model selection procedure results in similar yearly $r^2$ to the best models
but increases in both %MAE and %Bias are apparent with the selected models. Compared to the best models, yearly
mean %MAE for selected models increases between 3% and 18% and yearly absolute %Bias increases between 3%
and 37%. The years 2013, 2014, and 2015 yielded increases in %MAE of 18%, 17%, and 16%, respectively for
selected versus best models. In 2016 selected model %MAE exhibited an increase of only 3%. The absolute %Bias
increases 3% in 2013, 6% in 2014, 37% in 2015, and 24% in 2016 for selected versus best models.

**Table 4. Yearly and overall prediction errors of PHV-FSCA for best transferred models and the selected models. Best**
**transferred model errors involved fitting a model to all the data on each date and using these models to predict SWE on**
**dates in other years. Only the best model date-simulation date pair is considered. The selected model errors are derived**
**from the same ensemble of model date-simulation date pairs, but the model is selected based on the pillow SWE similarity**
**described in the text.**

|  | Best Transferred Model | | | Selected Model | | |
|---|---|---|---|---|---|---|
|  | $r^2$ | %MAE | %Bias | $r^2$ | %MAE | %Bias |
| 2013 | 0.86 | 34 | -1 | 0.84 | 52 | 4 |
| 2014 | 0.86 | 26 | 0 | 0.84 | 43 | 6 |
| 2015 | 0.82 | 39 | 3 | 0.81 | 55 | 40 |
| 2016 | 0.82 | 40 | -1 | 0.81 | 43 | -25 |
| **Mean** | **0.85** | **33** | **1** | **0.83** | **48** | **11** |



Figure 6 shows the difference for each transfer date between the errors of the best models (diamonds in Figs.s

4, 5) and the errors of the selected models (these are shown as open circles in Fig. 6). We focus on %MAE and %Bias
from PHV-FSCA only because $r^2$ showed generally consistent performance for a given transfer date (Table 4;
comparatively small vertical range of the purple boxplots in Fig. 3 compared with Figs. 4 and 5).





The mean difference in %MAE between the selected and best models is 15% and the selected model exhibited
the same %MAE as the best model on only one date (Fig. 6a). The range in performance difference is between 0%
and 45% with a mean of 15% and median of 12%. The differences in %MAE were generally lowest in 2016 with a
mean of 3% and standard deviation of 4%. In contrast, 2013, 2014, and 2015 exhibited both higher mean differences
(18%, 16%, and 18%, respectively) and higher standard deviations in %MAE (14%, 8%, and 14%, respectively).
The best model had a lower absolute magnitude %Bias of between 0% and 73% with a mean of 29% and
median of 26%. The selected model was the same as the best model on only one date (Fig. 6b). The yearly mean
difference in error was consistently higher for %Bias than %MAE, with means of 25% in 2013, 30% in 2014, 34% in
2015, and 20% in 2016. The standard deviation in the error difference was lowest in 2014 (9%) compared to 16% in
2013, 22% in 2015, and 11% in 2016.

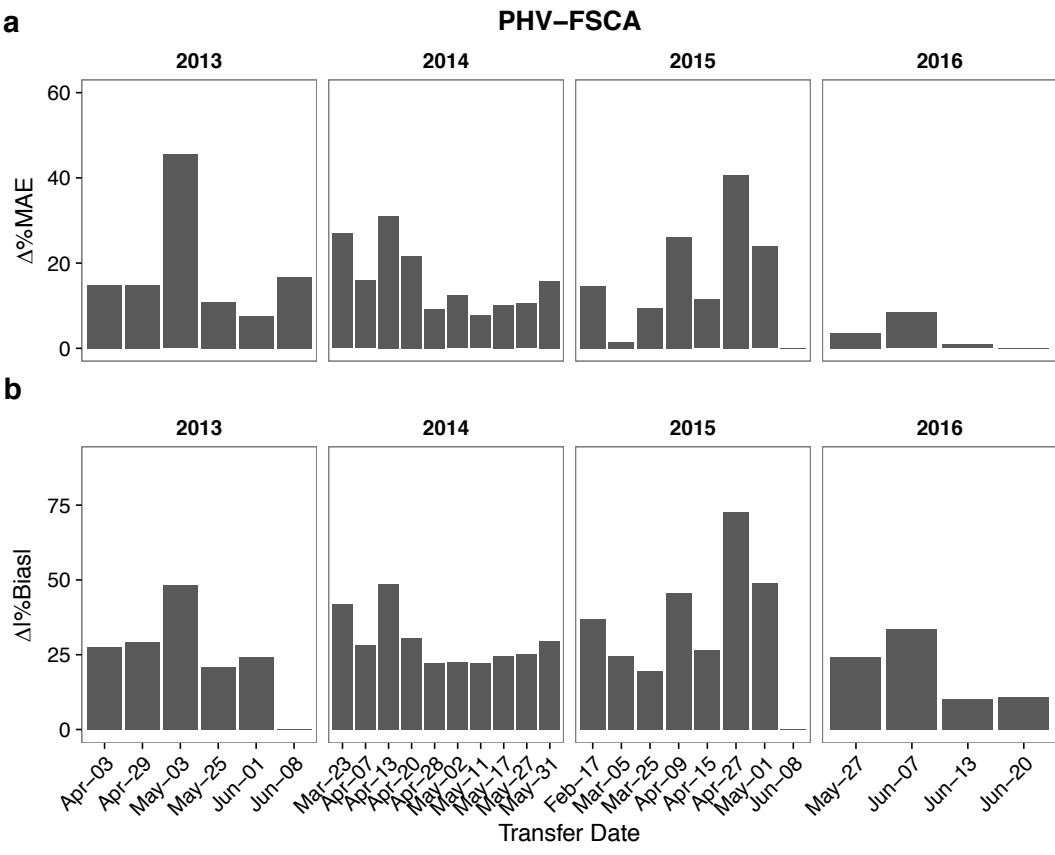


**Figure 6. The increase in prediction error for %MAE (a) and %Bias (b) between the best model and the selected model.**

We evaluate the prediction errors from different model years to see if there are any systematic differences in
the predictions generated; i.e. do some years yield better predictions of other years? This allows us to determine how
sensitive predictions for transfer dates will be given the existing ensemble of observations. In this context, we note





distinct differences in the predictive ability of models from different years, which has important implications for the
ability of a model to transfer in time.
Figure 7a shows that, on average, models *from* 2015 produced the lowest %MAE and models from 2016
produce predictions with the largest %MAE. The mean %MAE in 2016 is 118% compared to 79% with 2013 models,
69% with 2014 models and 50% with 2015 models. We observe consistency in a year's ability to predict another year
relative to the overall distribution, i.e. the colored dots are typically clustered within the range of the boxplots. We
also note in 2016 that the best models always came from 2013, but in 2013 the best models only came from 2016 for
the first three flights. In 2014, the bulk of the models around the median performance were from 2015 and vice-versa
in 2015. In these two years, the poorest predictions were from 2013 and 2016.
Figure 7b show that models *from* 2016 also produce the largest %Bias with a mean of 97% compared with
56% from 2013, 21% from 2014, and -26% from 2015. We again see consistency in the location of the colored dots
relative to the boxplots thus suggesting years will consistently model the SWE distribution of certain other years
better. In this regard, similar to %MAE, 2013 produces the lowest %Bias in 2016, but in 2013 the inverse is only true
for the first three flights as the distribution shifts up.

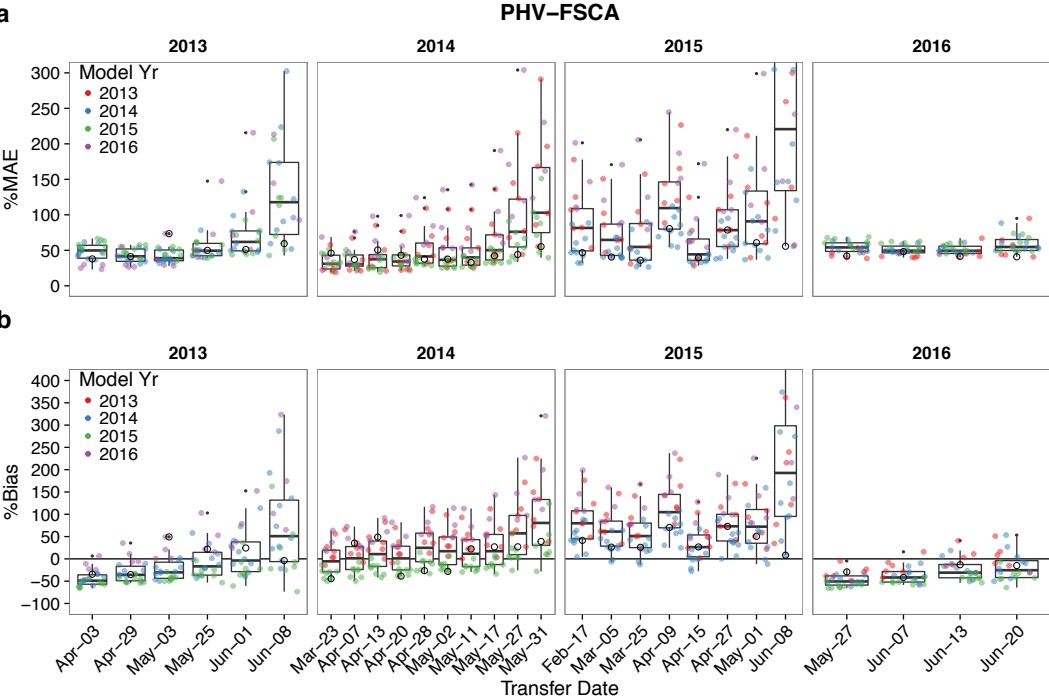


**Figure 7. The prediction errors for PHV-FSCA for each date; %MAE (a) and %Bias (b). The colored dots are the prediction**
**errors coded by model year and jittered to prevent over plotting. Open circles represent the selected model. The boxplots**
**represent the entire distribution of prediction errors for each date. The boxes represent the interquartile range with vertical**
**bars for the 5th and 95th percentiles. Small black dots are statistical outliers.**





## 6 Discussion

### 6.1 SWE Distribution Modeling

The relationship between snow covered area and SWE is well established as evident by the common use of depletion curves in hydrologic modeling (Anderson, 1973; Clark et al., 2011; Liston, 1999; Luce et al., 1999). We invert the idea of the depletion curve in this study by using the spatial distribution of fSCA to predict the spatial distribution of SWE. The basis for this approach is relatively well established given that fSCA is sensitive to topographic complexity and the spatial distribution of SWE (Donald et al., 1995; Fassnacht et al., 2016; Niu and Yang, 2007). In this regard, model performance consistently improved when fSCA was used as a predictor, with an average %MAE decrease from 61% with PHV to 32% with PHV-FSCA. The utility of fSCA as an explainer of SWE distribution is not an unprecedented result. König and Sturm (1998) showed that in situ measurements of SWE were correlated with fSCA from aerial photographs. In addition, Marchand et al. (2005) suggested that the sub-grid standard deviation of physiography could improve regression models of SWE distribution because it accounts for sub-grid snow depth variability. Remotely sensed fSCA provides a means of capturing this sub-grid variability in SWE without requiring higher resolution data to compute the variance of physiography for each pixel.

The statistics for the selected PHV-FSCA model reported in this study compare favorably to SWE distribution statistics reported previously. Headwater catchment scale studies, based on intensive field data, have been able to achieve $r^2$ values from as low as 0.18 to as high as 0.65 (Balk and Elder, 2000; Elder et al., 1991; 1998; Erxleben et al., 2002; López-Moreno and Nogués-Bravo, 2006; Molotch and Bales, 2005) compared to the mean $r^2$ of 0.83 for the selected models in this study. These papers, which cover only a few square kilometers, represent a far more simplistic problem with regard to characterizing relationships between snow accumulation and physiographic variables.

The results presented herein compare favorably with larger scale studies (i.e. > 1000 km$^2$) of snow distribution. Fassnacht et al. (2003) reported average yearly RMSE between 0.12 and 0.16 m and 0 m bias when cross-validated with snow pillows. Harshburger et al. (2010) reported an average $r^2$ of 0.82 and RMSE of 0.05 m when cross-validated with snow pillows. Schneider and Molotch (2016) reported a mean RMSE of 0.23 m and %Bias of 0.8% from snow surveys in the Upper Colorado River Basin. This is compared to a mean RMSE of 0.07 m and mean %Bias of 11% for the selected model in this study. The favorable error statistics reported here are even more encouraging when considering the differences in evaluation methods of these previous studies. In this regard, the error values reported here are quite robust in that we compare against spatially explicit observations over relatively large spatial scales (i.e. > 1000 km$^2$). In contrast, the aforementioned works were evaluated against relatively sparse observations (Fassnacht et al., 2003; Harshburger et al., 2010; Schneider and Molotch, 2016).

Bair et al. (2016) compared a retrospective SWE reconstruction to the same ASO observations (2013-2015) and reported yearly mean %MAE between 20% and 31% and yearly mean %Bias between -11% and 10%. These yearly statistics are better than those reported in this study (Tables 3, 4), but are the result of a much more complicated energy balance model that can only be run after the snow has disappeared. The selected model in this study is a simple linear regression that can be applied in real-time, thus we consider our results valuable for applications where real-time estimates of SWE distribution are needed. Furthermore, we compare our selected model results to SWE estimates



from the U.S. National Weather Service's operational Snow Data Assimilation System (SNODAS). SNODAS
produces spatially distributed SWE estimates for the coterminous United States at 1 km by assimilating a physically
based model with SNOTEL observations and remotely sensed snow covered area. The SWE product from SNODAS
is the only high-resolution, gridded SWE product available at a daily time step for the continental United States and
is available from http://nsidc.org (Barrett, 2003). Previous work has shown the physically based model to perform
well at the point scale (Rutter et al., 2008) but suffer in alpine zones because it does not consider wind redistribution
(Clow et al., 2012). The yearly mean $r^2$ between ASO and SNODAS ranges from 0.04 in 2016 to 0.36 in 2015 with a
mean of 0.17. The yearly mean %MAE ranges from 120% in 2014 to 274% in 2013 with a mean of 199%. The yearly
mean %Bias ranges from -10% in 2016 to 236% in 2015. We refer the reader to "Selected Model" of Table 4 for the
PHV-FSCA error summary. In this regard, SNODAS exhibits a mean %MAE 4 times greater than that of PHV-FSCA
and a mean %Bias 8 times higher than PHV-FSCA. Both models poorly predicted the anomalous conditions in 2015.
While it is clear that the errors with PHV-FSCA are considerably lower than with SNODAS overall, SNODAS is a
complex system that attempts to capture the snow dynamics across the entire United States compared with PHV-
FSCA which was trained using a very specialized data set in the study region.

We also show that SWE distributions can be related to fSCA and physiography in one year and applied to

another year. The performance of PHV-FSCA was quite similar when applied to the date at which the model was
trained (i.e. discrete time models with a split sample) versus applying the model to other years (i.e. transferred models).
In this regard, we see a minimal decrease in prediction skill and minimal increase in prediction error when we compare
split sample models with the best transferred models. Recall that the split sample model was trained and tested on the
same day and the transferred model was trained in a different year from which the model was applied. Table 3 shows
the mean $r^2$ in the split sample model to be equivalent to that of the best transferred model. Moreover, the average
%MAE of the best transferred model exhibits only a 1% difference from the split sample model and the mean %Bias
exhibits no difference. Thus, if we are able to identify the best model for a given date of interest we would see minimal
degradation in predictive ability relative to a model derived from data acquired on the date of interest. However, we
see significant differences in predictive ability from models of different years (Fig. 7) and conclude that relationships
between SWE and physiography are only similar between specific years, not as uniformly as put forth by previous
studies (Erickson et al., 2005; Grünewald et al., 2013). Also, it is unclear as to the impact of climate non-stationarity
with respect to the ability to transfer models to future years. Even so, for each year in this dataset there exists a
corresponding year from which accurate predictions can be made.

The benefits of using fSCA as a predictor variable in the transferred models are particularly large at the end

of the season when the errors are highest (Figs. 3, 4, 5). For the last 2 dates of each year, the average difference in
%MAE between the best PHV model and best PHV-FSCA model was 54% compared to an average difference of 20%
for the other dates. The improvements seen by including fSCA as a predictor are noteworthy because the ensemble of
models trained using ASO observations could then be applied using remotely sensed fSCA from satellites. The degree
to which satellite-based fSCA will improve model performance toward the end of the snowmelt season will be partially
dictated by the accuracy of the fSCA data, which is subject to increasing uncertainty at low fSCA values (Painter et
al., 2009; Rittger et al., 2013). Optical fSCA products such as MODSCAG also suffer in forested areas since snow



cover is occluded by the canopy (Raleigh et al., 2013; Rittger et al., 2013). A viewable gap fraction correction is
typically used to extrapolate fSCA to the occluded portions of a pixel by assuming that fSCA is the same under the
canopy as it is in canopy gaps (Molotch and Margulis, 2008), and future studies should evaluate the sensitivity of
PHV-FSCA to this assumption. Cloud cover can also obscure a satellite's view of the snow and therefore making it
difficult to estimate the snow extent. However, Slater et al. (2013) showed that gaps of 5 or more consecutive days
are rare with MODIS thus suggesting that weekly estimates of SWE would be feasible. Lastly, omission and
commission errors of cloud identification can provide erroneous estimates of fSCA although Parajka and Blöschl
(2008) report an overall filtering accuracy of 96% in the Alps. Nonetheless, this is still an active area of research
(Dozier et al., 2008; Parajka and Bloeschl, 2008; Rittger et al., 2013; Xia et al., 2012).
### 6.2    Considerations for Extending the ASO Record
The value of ASO during the year flown is significant for water management because it provides high resolution SWE
information with low uncertainty compared to traditional estimation methods and therefore facilitates more accurate
water supply forecasts. The downside to ASO is the relatively high cost of operation compared to traditional
measurement campaigns. The work presented here provides a first step in realizing the value of ASO subsequent to
active operations. We find that the number of flights within a year affects the mean and variance of the predictions in
other years by <1%; this was quantified by iteratively selecting between 1 and the number of observations in a given
year 100 times and assessing the change in error. In other words, it is better to perform ASO once per year for 10 years
than 10 times in one year.
It is clear from our results that flights in one year do not necessarily transfer well to another year, e.g. 2016 to
2014 and 2016 to 2015 (Fig. 7). The model selection procedure relies on operational snow pillows to identify similar
patterns of SWE between historical dates (i.e. model dates) and the date of interest (i.e. transfer dates). The spatial
representativeness of these stations may have contributed to the general inability of these stations to select the best
historical date for a given date of interest. In this context, it is well established that these snow pillows may not
adequately represent the SWE of the surrounding terrain (Meromy et al., 2012; Molotch and Bales, 2006; Rice et al.,
2011). However, the minimal degradation in prediction performance when considering the best transferred model
should motivate improvements for identifying the dates from the past with the most similar SWE distribution. As
shown here, if these dates can be identified, the SWE can be modeled at accuracy levels that are equivalent to models
generated from ASO data collected on the date of interest. We also tested the similarity in remotely sensed fSCA as a
method for identifying the historical date with the most similar SWE distribution to the date of interest. We found that
anomalous SWE and fSCA distributions make this method less robust than using snow pillow data. However, a
completely remote sensing based approach would be useful in data sparse regions where ground stations do not exist.
A potentially robust remote sensing based method could be to track fSCA through time and select a similar SWE
distribution based on the trajectory of fSCA rather than a single snapshot of fSCA. It is also important to note that
flying ASO during a year with an anomalously low snowpack such as in 2015 in California does not necessarily reduce
the predictive capacity of models for future years. In our case, 2015 provided useful estimates of SWE distribution in



all the other years, including 2016, even though the models from the relatively wet year of 2016 did not transfer well
to the very dry year of 2015 (Fig. 7).
Intuitively, it is not surprising that the models from 2016 did not transfer to other years because there was much
more snow than in 2014 and 2015, which were very low snow years. Less intuitive, however, is why the inverse
worked well, i.e. the models from 2014 and 2015 did transfer relatively well to 2016. In this regard, we see lower
mean fSCA in 2016 for the same mean SWE in 2015 in the ASO data. This means that for the same mean SWE, there
are deeper pockets of snow covering a smaller area remaining at the end of a higher snow year (i.e. 2016) than a lower
snow year (i.e. 2015). Therefore, we would expect the relationships between topographic variables and SWE to
become increasingly disparate from the underlying terrain with a deeper snow accumulation. Thus, the regression
coefficients derived from years with deeper snowpacks do not adequately represent relationships found during years
with shallow snowpacks. However, this does not mean that the model from the last date of the season (which is also
a shallow snowpack) transfers well because this SWE distribution still largely represents the dominant spatial patterns
from the peak SWE distribution (Egli and Jonas, 2009; Liston, 1999; Luce et al., 1999).

## 7    Conclusion

We estimated the relationships between SWE, physiography, and fSCA and show that the temporal consistency in
these relationships can be used to estimate SWE in years beyond the ASO observation record, with a mean $r^2$ of .85,
mean %MAE of 33%, and mean %Bias of 1%. The relationships transfer robustly in time with no degradation in $r^2$ or
%Bias and only 1% in %MAE when comparing predictions between models fit on the same day and models from a
different year. Models with fSCA as a predictor transfer better than those without, and we suggest that the inclusion
of fSCA provides information with regard to the variability of the SWE resulting from different accumulation
dynamics due to differences in terrain roughness. In this regard, the availability of satellite images of fSCA facilitate
the transfer of modeled relationships based on ASO observations to dates when no airborne snow depth measurements
exist. The crux of this proposition is in selecting the model to transfer. We offer a method with which to select a model
from another year based on the similarity in SWE distribution at existing snow pillows in the area. Comparison of the
best predictions and the selected predictions results in a mild decrease in $r^2$ (0.02) and moderate increases in %MAE
(15%) and %Bias (10%). The results presented above motivate further refinement in the technique used to select the
best model because if these dates can be identified then SWE can be modeled at accuracy levels equivalent to models
generated from ASO data collected on the day of interest. Lastly, although SWE distributions simulated in years with
anomalous SWE distributions (2014, 2015) had the highest errors, models from these years still yielded good
performance in 2013 and 2016. Thus, the benefit of ASO in anomalously dry years is two-fold: water managers receive
accurate information during a year that is difficult to model, but also these observed SWE distributions can be used to
simulate SWE distributions in future, less anomalous years. Overall, ASO provides an unprecedented observation of
the relationships between SWE, fSCA, and physiography. The ASO dataset facilitates improved understanding of
these relationships in both time and space and should lead to better information for water managers.



## 8 Data Availability

All data is available from the corresponding author at noah.molotch@colorado.edu.

## 9 Acknowledgements

D. Schneider thanks his PhD committee for their insightful comments. The authors would like to acknowledge NASA for supporting D. Schneider with an Earth and Space Science Fellowship (NNX14AL27H) and funding the time of N.P. Molotch with grant numbers NNX17AF50G and 80NSSC17K0071.

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
