# Peer review of "Estimating relationships between snow water equivalent, snow covered area, and topography to extend the Airborne Snow"

_The Cryosphere, 2017_

## Referee Comment (RC1) · Anonymous Referee #1 · 6 Nov 2017

General comments

The authors present a comprehensive study on how Snow Water Equivalent distribution in mountain areas can be derived from terrain topographic characteristic, vegetation height and fractional snow cover data from satellites. The methodology is tested with a high quality database form the Airborne Snow Observatory. The spatio-temporal extension of this database over Toulumne Basin provides a unique opportunity to test the methodology. Results obtained from the statistical approach they describe, show the high potential of the methodology. As far as I know, this work presents the first method intended to obtain SWE distribution combining satellite information and terrain

features over extended areas, showing quite promising results. The article is well written, with methods and results in general adequately articulated. The database and the methodology described are suitable for the field of research. Results are sufficiently discussed with suitable references included. The article also suites the scope of the journal, introducing a novel approach which may be applied in other mountain areas with feasibly good results. However, I consider some issues must be clarified and also some complementary hypothesis could be tested, in order to present a more compelling methodology. Thus I recommend its publication after moderate changes described here in after.

Major points to be included/discussed:

1. Necessity to include more information for some methods/databases presented in the manuscript. SWE distribution maps are generated from snow depth information directly obtained from ASO flights (elevation difference between snow and free snow acquisitions) and an energy balance model as it is stated from lines 55-56. Painter et al. 2016, describe the methodology to generate SWE distribution maps providing detailed information. Nevertheless since this database is the main observation to adjust the linear regression models and also to test the methodology, I encourage manuscript authors to provide more information about how density is simulated, which in-situ data are integrated (are snow pillows observations included?) and how the final SWE product is generated.

2. Since SWE information is available on a spatial resolution of 50 m, the vegetation height in a 3 m resolution and terrain topographic variables could be derived from a high spatial resolution DEM (I guess in this study area a 3 DEM may be available), exploiting fSCA information; the statistical models could be intended to generate same spatial resolution of ASO final SWE product. Has been tested this hypothesis?, does manuscript authors planned to do so? Argue in discussion section future work regarding the latest advances on deriving snow absence/presence from fSCA and terrain topographic characteristics (Cistera et al., 2017) to increase the spatial resolution of

simulated SWE distribution.

3. In section 4,2, it is stated that models obtained in a particular year are not used to obtain SWE distribution in other dates of the same year. I understand that authors want to test models without including any information about SWE distribution in a particular year. However, from my point of view, having some information about SWE distribution on a particular date could be quite interesting for many applications and may reduce the uncertainty on determining SWE. Indeed, somehow, in the discussion this idea is supported (lines 486-488). This way if SWE observations obtained on early snow season are used to generate a model for the same year, it could be more accurate on describing SWE distribution within this snow season. I encourage manuscript authors to explore this hypothesis in methodology and result sections and also to argue about it on discussion section.

4. Some topographic variables are not sufficiently explained. For instance the "Vector ruggedness measure" and the "Topographic Position Index" both of them have a high dependence on the searching distance. Could you please specify if you have performed preliminary analysis to determine which is the best searching distance within your study area or if you have used values from bibliography? Please, include more details on how these variables have been obtained; this will help potential readers to apply the methods described in different study areas.

Minor comments

Line 17: Move performance values, in brackets; to lines 21-22 to show the final performance after showing the mid decrease between the best model and the selected one.

Lines 33- 34: Add one or two references that exemplify the poor data availability of SWE observations over large areas (i.e. SNOTEL program)

Line 35-36: Cite: López-Moreno, J. I., Fassnacht, S. R., Heath, J. T., Musselman, K. N.,

Revuelto, J., Latron, J., Morán-Tejeda, E., and Jonas, T. Small scale spatial variability of snow density and depth over complex alpine terrain: Implications for estimating snow water equivalent, Adv. Water Resour., 55, 40–52, 2012

Line 42: Cite: Prokop, A.: Assessing the applicability of terrestrial laser scanning for spatial snow depth measurements, Cold Reg. Sci. Technol., 54, 155–163, 2008

Line 55: Please cite which energy-balance model.

Line 64: Here and throughout the whole manuscript; I encourage changing "physiography" by "topography" and "vegetation height" since it is often used on snow literature.

Line 74: Cite: Revuelto J, López-Moreno JI, Azorin-Molina C, Vicente-Serrano SM. 20014b. Topographic control of snowpack distribution in a small catchment in the central Spanish Pyrenees: intra- and inter-annual persistence. The Cryosphere 8(5): 1989–2006. DOI:http://dx.doi.org/ 10.5194/tc-8-1989-2014.

Line 76-78: Remove or merge sentence with lines 43-45.

Line 85-86: Please include here some information about the spatial extent of ASO; time duration and total number of observations available.

Line 90: Include other more recent citations such as: Molotch N.P. and Margulis S.A. 2008: Estimating the distribution of snow water equivalent using remote sensed data and a spatially distributed snowmelt model: a multi-resolution, multi-sensor comparison. Advances in Water Resources Research, 13-1503-1514

Line 98: Include these two references. Sturm M, Wagner AM. 2010. Using repeated patterns in snow distribution modeling: an Arctic example. Water Resources Research 46, W12549. DOI: http://dx.doi.org/10.1029/2010WR009434, and Revuelto, J., Jonas, T., & López-Moreno, J. I. (2016). Backward snow depth reconstruction at high spatial resolution based on time-lapse photography. Hydrological Processes, 30(17), 2976-2990.

Line 117: You say "...unsampled dates of interest". However dates considered here have been sampled by the ASO. Please rephrase.

Line 122-Figure 1: Add in this figure an extra map with a DEM showing the heterogeneity of the study area. This information will help to interpret SWE distribution shown in Figure 2. Is there any ice body in the study area? If it is, in the land cover map it is really difficult to see, please highlight it or remove this land cover class from the legend.

Line 124 - 127: Please, include scientific names of the trees occurring in the different forests types.

Study area section: Climatic information of the study area is highly desirable to understand the importance of snowpack evolution in this site. Please, include a paragraph describing the main climatic characteristics and also the average snowpack values observed during the study years. This information will show the characteristics of the analyzed years (below average snowpack conditions).

Line 139-140: How snow depth and SWE maps are generated? Why these do not have same spatial resolution? I encourage manuscript authors to include here more information about the model used to generate SWE products from snow depth.

Table 1: Source column for VRM, Lopez et al. (2014) should be Lopez-Moreno et al. (2014).

Line 180: "discrete points of time" could be confusing. Please reword to make clear it is only considering specific dates.

Line 188: I guess you mean you split the SWE distribution data from each ASO flight. Please clarify in the text.

Line 187-189: In subsequent sections you compare "split-sample models" with results obtained from "best transferred model". Please present in these lines "split-sample models" similarly as you do later with "best model" or "transferred model".

[Figure]

Line 217: When you talk about nearby snow pillows, you consider only these of Figure 1? all of them?, do you take into account their distance to the study area? their elevation? Please clarify all these questions.

Line 222: "best model" on italics Please, maintain this criteria along the text. This is, when you talk about "best model", "selected model", "split mode"..., do it always in italics since it is a convention adopted in methods section.

Line 252: Change "alpha" by the Greek symbol as presented before.

Figure 2. It is highly desirable to include elevation contour lines on SWE maps and also in the difference maps to better interpret results.

Line 284-287 A reduced map with Land Cover information included on Figure 2 could really help to interpret results.

Line 288: You can also say that in these pixels you observed the higher SWE values (at least on 2014-03-23). Indeed, since Google Earth information is not a database exploited in this article, I suggest removing this sentence and state that these pixels correspond to areas in which you have observed high SWE values from ASO data and also long snow presence from MODIS data.

Figure 3, (4, 5): Since you present box plots of r2 values for 18 to 24 potential models for each date (depending on the year), to me, it is not needed to include diamonds showing the best model, because it is the 95th (or the 5th, depending on the figure)percentile in all cases. Thereby I would remove it to show graphs easier to interpret. If finally you decide to maintain diamonds, include it on the legend of the figures.

Line 343: Remember here that "Selected models" are those selected from the similarity on snow snow pillows observations.

Line 358: I guess you mean Figures 3, 4 and 5 and not Figs.s,4,5.

Line 359: I guess you meant open circles in Figure 7.

Line 363: If I am right, figure 6a shows two dates with same %MAE, one in 2015 and one in 2016 (both for the last ASO observation). Please verify this and include this appropriately in the text.

Figure 7: Please, change open circles by a different symbol because in some cases it is difficult to see their position. Please include this symbol in the legend.

Line 404: I suggest starting this section talking about snowpack and citing other recent works that also exploit fSCA such as: Cristea et al., 2017 and Walters et al., 2014. Afterwards, you can talk about SWE distribution as you do. This will show that many researches are working on combining the influence of topography on snowpack distribution with satellite observations with different approaches.

Line 416-416: I consider it is a bit presumptuous to say that "these papers, which cover only a few square kilometers, represent a far more simplistic problem". Some of the works you cite in the previous sentence cover large areas and others, despite the smaller extension, have a different spatial scale, data base etc,....Please remove this sentence, you have already shown that your methods performs well and are pushing forward snow science.

Line 437: I guess SNOTEL observations correspond to SWE data from snow pillows, please include before this information in the text (Figure 1, 4.2 section when you talk about "selected model"...)

Line 434: I guess you meant you are going to present the comparison between SNODAS and PHV-FSCA results in this section. Please rephrase, it could be confusing this sentence.

Line 442: When you state "the yearly mean", it is form results obtained in this study or from SNODAS-ASO comparison? Please clarify.

Line 499-500: When did you exactly perform the similarity in remote sensed fSCA? Please introduce it before in methods section, when you talk about similar SWE distribution from snow pillows.

Conclusion section: Since at the end of the introduction section you state three main questions, I suggest to directly including the question and their answer in the conclusions. The answer is somehow stated, but main conclusions will be more directly linked to questions previously stated.

References.

Cristea, N. C., Breckheimer, I., Raleigh, M. S., HilleRisLambers, J., & Lundquist, J. D. (2017). An evaluation of terrain-based downscaling of fractional snow covered area datasets based on Lidar derived snow data and orthoimagery. Water Resources Research.

Painter, T. H., Berisford, D. F., Boardman, J. W., Bormann, K. J., Deems, J. S., Gehrke, F., Hedrick, A., Joyce, Laidlaw, R., Marks, D., Mattmann, C., McGurk, B., Ramirez, P., Richardson, M., Skiles, S. M., Seidel, F Winstral, A.: The Airborne Snow Observatory: Fusion of scanning lidar, imaging spectrometer, and physically- based modeling for mapping snow water equivalent and snow albedo, Remote Sensing of Environment, 184, 139-152,

Walters, R. D., Watson, K. A., Marshall, H. P., McNamara, J. P., & Flores, A. N. (2014). A physiographic approach to downscaling fractional snow cover data in mountainous regions. Remote sensing of environment, 152, 413-425

---

## Referee Comment (RC2) · Anonymous Referee #2 · 28 Nov 2017

General Comments

This paper uses observational snow data collected as part of the Airborne Snow Observatory (ASO) in the Tuolumne Basin, California over multiple years and at different times of the melt period to examine temporal persistence in the patterns of snow water equivalent (SWE) and their relationship with various terrain parameters. Multiple regression analysis is used to model the relationship between SWE and the other parameters as independent variables–not from the original high resolution ASO products (3 m horizontal resolution), but from coarser spatially aggregated (to 500 m resolution) representations of these data. The main reported contribution is in using fractional

snow cover area (fSCA) information from the ASO dataset to improve the transferability of the regression models to other dates when the detailed ASO snow observations are lacking, with the premise being that other available fSCA products, such as from MODIS, could be used instead at these times.

While the purpose and rationale seem clear enough and there is value in using the exemplary dataset from ASO to develop better approaches for predicting SWE and snowcover patterns when detailed observations are not available, the study suffers from major conceptual and methodological flaws that greatly limit its practical usefulness. From a conceptual perspective, it makes little sense and it is not particularly useful to develop a set of different regression models for each of the ASO acquisition dates, and then to later determine which one to use based on some similarity criteria from snow pillow observations. It would be more useful, for example, to develop a single model or approach to characterize spatial patterns of SWE where observations such as SWE at the snow pillows and remotely sensed snow cover information are used in combination to inform this. Or more simply, why not just use the original ASO datasets themselves based on similarity with the snow pillow data? Of course the limits with this are clearly apparent. Given the empirical nature and lack of any incorporation of snow accumulation, redistribution, and ablation processes in the approach, the results are not likely to transfer well outside of this basin, and quite possibly not even to other years and seasons under altered climate conditions.

Even more concerning is the methodological approach. The ASO data provide an opportunity to explore patterns of SWE variability and their changes over time at a high level of detail, and yet the approach here has been to aggregate these data and lose that valuable information. The local scale patterns of SWE accumulation are no longer captured, which is the whole purpose of relating SWE to terrain parameters (i.e. the drifts form in sheltered locations while exposed sites are more wind scoured, and much of this variability occurs at scales from meters to 10s of meters in complex mountain terrain). Moreover, the terrain parameters that are derived from this aggregated DEM

become physically meaningless. How useful is average vegetation height over a 500 m grid? What do vector ruggedness or topographic position index really mean at this scale, for example? Certainly nothing from a physical sense that relates to patterns of SWE accumulation due to drifting, wind scouring, trapping of blowing snow by exposed vegetation, etc., and especially when relating to the averaged SWE over a 500 m grid.

In addition to these concerns, the approach itself seems quite overcomplicated for what is essentially just an examination of the temporal persistence in SWE patterns over time, and ultimately an attempt to determine which regression model best fits the conditions of a given time based on a set of in situ SWE observations. The methods are not well described and hard to follow in some parts, leaving some major doubts. I will explain more in my specific comments below.

In the end, the study makes only a very incremental contribution to this important topic. While the ASO data provide unique opportunities to explore relationships between SWE, terrain, and snow cover patterns, and to advance understanding of the process and terrain controls in a way that could lead to improved predictability of SWE patterns, this study takes this in a different direction and instead focuses on coarser resolution regression analyses that largely miss this. Although the authors argue that this approach is novel and no one has previously used fSCA in such a regression, there is no major theoretical advancement, no practical utility as a result of the flawed methodology, and the results are severely limited in their broader applicability elsewhere. I would therefore recommend this paper to be rejected.

Specific Comments

Page 5, 6 - Data Sources: The ASO data includes a 3 m snow-free DEM, 3m snow depth, 3 m vegetation height, and 50 m SWE information. From this it would be straightforward to estimate 3 m SWE, and this is the appropriate scale to work at in deriving relationships between snow cover, SWE, and terrain. It makes little or no sense to aggregate to 500 m. This greatly affects the subsequently derived terrain parameters and

alters the SWE distribution through averaging. The rationale in this study is to move to a scale where fSCA can be used as another variable in the analysis, but this is done at the cost of the detailed information on SWE patterns, which is key for such an analysis.

Page 7, lines 173-174: The "PHV" model is a multiple regression that uses only physiographic variables as independent variables. This is fine, but to use such an approach in a predictive sense would require other information to adjust the results for a particular time (i.e. SWE observations at snow pillows, other in situ data, etc.). What is instead done here is to develop a whole suite of regression models for the different ASO dates and cross-compare against the observations on other dates to pick which fits best.

Page 7, lines 178-179: If you are masking the regression estimates to the ASO observed snow cover areas, are you not greatly influencing the results and the reported performance of the model? Shouldn't areas without snow cover be included? This is not clear. And what about when the model is to be used in a predictive sense when there are no ASO observations?

Page 8, lines 281-222: To pick which model dates exhibits the greatest similarity with the transfer date, the mean error in SWE at snow pillows is used. As currently described in the paper, this is unclear. Is this error between the model predicted SWE (for 500 m grid squares) and the snow pillows? How would this be later used in a predictive sense? As I understand, to find the selected model, you need to compare the results of all of them against the snow pillow observations.

Page 15, lines 358-359: the text refers to selected models being shown as open circles in Fig. 6. There are no open circles in Fig. 6.

Page 18, lines 432-434: The selected model is a simple linear regression to be applied in real time. How? As noted above, this is not clear. Do you need to simply look at all of the models and see which fits best to the SWE at the snow pillows, and then choose that one to predict SWE patterns? How exactly is this helpful at all and how does it advance the science? For instance, if this is indeed the case, I would argue

that it would be far better to look at the original ASO data, with its fine scale detail and high accuracy, and simply compare that against the snow pillow observations for a given date. If all that is gained is to use fSCA as another parameter in the model for prediction, what about the point the authors make on page 20, lines 499-501 that there are anomalous SWE and fSCA distributions that limit the usefulness of using fSCA as an indicator for model selection? How are predictions to be made with any confidence in circumstances where the observed SWE (and perhaps fSCA) differs greatly from the previous conditions observed during ASO campaigns, and thus how can these results extend the ASO record as suggested? This is a fundamental weakness of the study.
* * *

---

## Author Comment (AC1) · 2 Mar 2018

We thank the editor and reviewers for their time and comments on our manuscript. We have responded to each comment in a response document (attached) and made changes to our manuscript accordingly (attached with and without tracked changes).

Please also note the supplement to this comment: https://www.the-cryosphere-discuss.net/tc-2017-167/tc-2017-167-AC1-supplement.zip